# WiFi Energy-Harvesting Antenna Inspired by the Resonant Magnetic Dipole Metamaterial

**DOI:** 10.3390/s22176523

**Published:** 2022-08-30

**Authors:** Zhenci Sun, Xiaoguang Zhao, Lingyun Zhang, Ziqi Mei, Han Zhong, Rui You, Wenshuai Lu, Zheng You, Jiahao Zhao

**Affiliations:** 1Department of Precision Instrument, Tsinghua University, Beijing 100084, China; 2State Key Laboratory of Precision Testing Technology and Instruments, Tsinghua University, Beijing 100084, China; 3Beijing Advanced Innovation Center for Integrated Circuits, Beijing 100084, China; 4School of Instrument Science and Opto-Electronic Engineering, Beijing Information Science and Technology University, Beijing 100016, China; 5Qiyuan Lab, Beijing 100089, China

**Keywords:** WiFi energy-harvesting, metamaterial, magnetic dipole, voltage multiplier

## Abstract

WiFi energy harvesting is a promising solution for powering microsensors and microsystems through collecting electromagnetic (EM) energies that exist everywhere in modern daily lives. In order to harvest EM energy, we proposed a metamaterial-inspired antenna (MIA) based on the resonant magnetic dipole operating in the WiFi bands. The MIA consists of two metallic split-ring resonators (SRRs), separated by an FR4 dielectric layer, in the broadside coupled configuration. The incident EM waves excite surface currents in the coupled SRRs, and the energy is oscillating between them due to near-field coupling. By varying the vertical distance of the two SRRs, we may achieve impedance matching without complicated matching networks. Collected EM energy can be converted to DC voltages via a rectifier circuit at the output of the coupling coil. Measured results demonstrate that the designed MIA may resonate at 2.4 GHz with a deep-subwavelength form factor (14 mm×14 mm×1.6 mm). The WiFi energy-harvesting capability of the proposed MIA with an embedded one-stage Dickson voltage multiplier has also been evaluated. A rectified DC voltage is approximately 500 mV when the MIA is placed at a distance of 2 cm from the WiFi transmit antenna with a 9 dBm transmitting power. The proposed compact MIA in this paper is of great importance for powering future distributed microsystems.

## 1. Introduction

With the rapid development of Internet-of-Things (IoT) applications and wireless sensor networks (WSNs), the number of microsystem nodes is exploding exponentially [1,2,3,4,5]. It is not possible to replace/recharge all their batteries [6] or power the microsystems by connecting electrical cables [7]. In order to meet the urgent demand for sustainable energy supply, energy-harvesting technologies were developed [8]. A large number of methods have been proposed to harvest ambient energy, such as wind energy [9,10], solar energy [11,12,13], thermal energy [14,15], radio frequency (RF) energy [16,17], and kinetic energy [18,19], among others.

Currently, WiFi systems are the most commonly used in wireless communication. Electromagnetic radiation operating in the WiFi bands is becoming ubiquitous and would be a potential environmental energy source [20,21,22]. If this RF energy can be efficiently harvested to power microsystems and microsensors, WSN nodes will operate in the self-power mode. Thus, we are able to save additional energy supplies to a large extent.

In recent years, harnessing WiFi energy is of great interest, and a lot of work has been conducted to harvest WiFi energy efficiently [23,24,25,26,27,28]. In general, the most commonly used system architecture of a WiFi energy-harvesting antenna (rectenna) consists of a receiver antenna, an impedance-matching network, an RF-DC rectifier, and other parts [16,29,30]. Among these components, the impedance of the system varies as a function of both input power and frequency due to the nonlinearity of electrical components in rectifiers [31,32]. The energy-harvesting capability of the system is also easily affected by rectifier circuits and loads [33]. Thus, the optimization process of the impedance matching circuit is complicated, and impedance mismatching dramatically decreases the conversion efficiency [34]. At present, most wireless energy-harvesting modules have been designed with conventional low-profile antennas [35]. Sun et al. proposed a 2.45 GHz rectenna based on a 87 mm × 80 mm two-element dipole array, which showed a maximum conversion efficiency of 83% with a load resistance of 1400 Ω [23]. Chandravanshi et al. developed a 2.45 GHz flexible rectenna system with a 92 mm × 70 mm dual-ring shaped monopole antenna to generate 0.98 V at a distance of 20 cm from the horn antenna with a 9 dBm transmitting power [24]. Wang et al. utilized a 31 mm × 18.5 mm patch antenna with a Koch fractal slot structure to harvest 2.42 GHz RF signals, and the highest conversion efficiency of the rectenna is 62% at the input power of 0 dBm [25]. Advanced microsystems and small wireless network nodes require miniaturized WiFi energy-harvesting units. However, the antenna size has become a major factor limiting the miniaturization of microsystems [36]. Therefore, the size of antennas needs to be reduced.

Metamaterials provide a design paradigm to achieve compact and electrically small antennas [37,38,39,40,41]. Metamaterials consist of subwavelength unit cells to obtain artificial electromagnetic properties, such as negative refractive indexes, zero refractive index, invisibility cloaking, and so forth [42,43,44,45]. As a classical metamaterial unit cell structure, split-ring resonators (SRRs) exhibit effective magnetic permeability due to the magnetic dipole-like response, which can generate loop currents under the excitation of the magnetic component of incident EM waves [46,47,48]. SRRs have been adopted as a design for electrically small antennas that coils up the conductors in a limited space [49]. For a pair of SRRs in the broadside coupled configuration, the near-field interaction will modify the resonant response of the single, constituent SRR and induce mode splitting [50,51,52]. Therefore, the antenna inspired by broadside coupled SRR may exhibit tunable impedance when adjusting the coupling strength between the two SRRs.

In this paper, we proposed a metamaterial-inspired antenna (MIA) that consists of broadside coupled SRRs. One of the SRRs acts as a resonant magnetic dipole, while the other one is a coupling coil that connects a rectifier circuit. The SRRs are separated by the FR4 substrate. We demonstrated an approach to tune the MIA impedance by adjusting the coupling factor between the SRRs. An equivalent circuit model (ECM) was proposed to explain the observed behavior. The overall volume of the MIA is 14 mm × 14 mm × 1.6 mm. The characteristic of energy harvesting is verified by both simulated and experimental results. Such an MIA with a single-stage voltage multiplier can supply power over 50 µW for a 1 KΩ load resistor at the 2 cm distance from the WiFi antenna, which enables an efficient WiFi energy harvester for low power consumption microsystems and microsensors. We present an endeavor to design an electrically small resonant antenna structure and optimize the RF-DC energy conversion circuit, expanding the practical applications of WiFi energy harvesters.

## 2. Design and Methods

### 2.1. Metamaterial-Inspired Antenna Design

The designed MIA consists of a resonant magnetic dipole (SRR1), coupling coil (SRR2), and FR4 substrate, as shown in Figure 1a. Both the resonant magnetic dipole and coupling coil are identical square SRRs. The two SRRs are placed in parallel on the upper and lower sides of the FR4 substrate, and their gaps are rotated 180° to each other, forming a broadside coupled configuration. As shown in Figure 1b, the resonant frequency of the MIA can be tuned to cover a relatively wide range in the WiFi band by changing the side length of the SRR.

When the EM wave is incident, the magnetic field component along the y-axis as shown in Figure 1a may excite oscillating current in the resonant magnetic dipole (SRR1). Strong enhancement of the electric field is confined at the gap area, as shown in Figure 1c. As shown in Figure 1d, surface currents can be excited in the coupling coil (SRR2) due to the strong near-field coupling between two SRRs. The induced oscillating current in SRR2 can be converted to the DC voltage utilizing the rectifier circuit for harvesting the wireless energy.

The electromagnetic responses of the MIA were studied by the numerical simulation. Figure 2a illustrates the schematic of the simulation model, which is composed of a WiFi transmitter antenna and an MIA. The transmitter antenna was modeled by an electric dipole antenna consisting of a single thin cylinder modeled as a perfect electric conducting (PEC) material with a total length of ≈ λ/2. A gap was formed in the PEC cylinder as port 1 for feeding the excitation. In the numerical model, the distance between the center of the MIA and the transmitter antenna is 10 cm. The SRRs are modeled by 50-μm-thick copper. The electric conductivity of copper is 5.8 × 10^7^ S·m^−1^, and the dielectric constant of the FR4 substrate is 4.3. The reflection coefficient of the MIA (S22) was simulated for various side lengths of SRRs. As shown in Figure 2b, the resonant frequency shift from 1.98 to 2.8 GHz as the side length decreases from 9 to 13 mm. At the same time, the minimum value of the MIA reflection coefficient (S22) fluctuates within a range of less than 0.15. The design parameters of the optimal MIA with a resonant frequency of about 2.4 GHz are shown in Table 1.

In addition to the side length, the gap of SRRs is another parameter that affects the resonant frequency. As depicted in Figure 2c, the resonant frequency increase from 2.34 to 2.49 GHz when the gap of SRR is changed from 0.4 to 1.6 mm. The reflection amplitude at the resonant frequencies is almost constant for various gap sizes, indicating that the gap has little effects on the MIA impedance at the resonant mode. Thus, the resonant frequency can be finely tuned without influencing the amplitude of S22.

By changing the thickness of the FR4 substrate (*t*_2_) with a 0.1 mm incremental step, we can investigate the effect of the interaction between the coupling coil and the magnetic dipole, as shown in Figure 2e,f. It can be seen that the minimum value of S22 first decreases to 0.01 and then increases, as *t*_2_ increases dramatically from 1 to 6 mm. An optimal impedance match is achieved when *t*_2_ = 3.1 mm. The change is mainly due to the fact that the coupling strength of the resonant magnetic dipole and coupling coil is affected by *t*_2_ [53].

### 2.2. MIA Modeling

The equivalent circuit model (ECM) of the MIA, as shown in Figure 3a, was built and employed to analyze the MIA response. The magnetic dipole and coupling coil can be simplified as a circuit with a serially connected resistor, inductor, and capacitor. All electrical parameters are assumed to be identical in two since they have identical SRR structures, i.e., *R*_1_ = *R*_2_ = *R*, *L*_1_ = *L*_2_ = *L*, and *C*_1_ = *C*_2_ = *C*. The self-inductance *L* of a metallic SRR can be calculated by the Neumann model [54], which is
(1)$$L=\frac{\mu_{0}l}{4\pi }\left\{2sinh^{-1}\left(\frac{l}{w}\right)+2\left(\frac{l}{w}\right)sinh^{-1}\left(\frac{w}{l}\right)+\frac{2}{3}\left[{\left(\frac{l}{w}\right)}^{2}+\frac{w}{l}\;–\;\frac{{\left(l^{2}+w^{2}\right)}^{\frac{3}{2}}}{lw^{2}}\right]\right\}$$
where $\mu_{0}$ is the permeability of the vacuum, *l* is the total length of the SRR (*l* = 4*l*_2_ − *gap*), and *w* is the linewidth of the SRR.

The mutual inductance $M_{12}$ between the inductor may be calculated by:(2)$$M_{12}=k\sqrt{L_{1}L_{2}}=kL$$
where *k* represents the coupling coefficient. The coupling coefficient can be tuned by varying the distance between *L*_1_ and *L*_2_.

Equations (3) and (4) can be obtained based on Kirchhoff’s law [55]
(3)$$j\omega M_{12}i_{2}+\left(R_{1}+j\omega L_{1}\right)i_{1}=V_{1}$$
(4)$$\left(R_{2}+j\omega L_{2}+\frac{1}{j\omega C_{2}}\right)i_{2}+j\omega M_{12}i_{1}=0$$
where *ω* is the operation frequency in radians, *i*_1_ and *i*_2_ are the currents inside the magnetic dipole and coupling coil, respectively, and *V*_1_ is the voltage across the magnetic dipole.

The input impedance $Z_{in}$ of the MIA can be derived by solving the above equations [55]; that is
(5)$$Z_{in}=\frac{R-2\omega^{2}RCL+j\omega \left[R^{2}C+L-\omega^{2}L^{2}C\left(1-k^{2}\right)\right]}{1-\omega^{2}\left(2LC+R^{2}C^{2}\right)+\omega^{4}L^{2}C^{2}\left(1-k^{2}\right)+j\omega \left(2RC-2\omega^{2}RLC^{2}\right)}\;$$

The reflection coefficient *r* can be calculated by [56]
(6)$$r=\left|\frac{Z_{in}-R_{s}}{Z_{in}+R_{s}}\right|$$
where $R_{s}$ is the source impedance equal to 50 Ω. The calculated reflection coefficient may fit the simulation result well, indicating that the ECM is sufficiently accurate to model the MIA response. For our designed MIA, self-inductances *L* were 46.216 nH calculated by Equation (1). Then, the values of the circuitry parameters were retrieved from fitting the simulated reflection coefficient by Equations (5) and (6). Calculated resistances *R*, capacitances *C*, and coupling coefficient *k* were 10.46 Ω, 0.1266 pF, and 0.5047, respectively. As shown in Figure 3b, the increase in capacitances (0.115–0.135 pF) contributes to a near-linear increase in resonance frequency without changing the value of the *r* dip (≈0.01). Thus, the proposed ECM can be used to explain the influence of gap on the reflection coefficient shown in Figure 2c.

To further reveal the relationship between the MIA reflection response and the near-field coupling of two SRRs, the reflection coefficients for various coupling coefficients *k* are investigated using the ECM, as shown in Figure 3c. As the coupling coefficient increases from 0 to 1, the resonant frequency blueshifts, and the reflection coefficient at the resonant frequency, or namely the *r* dip, reaches a minimum at a specific coupling coefficient, showing a similar trend with Figure 2e. As shown in Figure 3d, the MIA reflection spectra with three different coupling coefficients (*k* = 0.2, *k* = 0.5047, and *k* = 0.65) clearly demonstrate the shift of the line shape. The value of *r* dip is minimum and approaching zero for the critical coupling coefficient, i.e., *k* = 0.5047, indicating a perfect impedance match. When the coupling coil and magnetic dipole are weakly coupled (*k* = 0.2) and strongly coupled (*k* = 0.65), the minimum value of *r* dip becomes larger. Therefore, impedance matching can be achieved by optimizing the distance of two SRRs, which is related to the coupling coefficient *k*.

### 2.3. System-Level Simulation of the Energy Harvester

The EM energy-harvesting system is simulated using the numerical model in connection with the SPICE model in CST Microwave Studio, as shown in Figure 4a. A sinusoidal signal with a tunable resonant frequency (2–2.8 GHz) is fed to the excitation port in order to mimic the WiFi source. The wireless energy is transmitted from the dipole antenna, and the energy is harvested by the MIA.

In addition, a single-stage Dickson voltage multiplier was integrated with the coupling coil to obtain a direct current (DC) voltage via the rectification effect [57]. Herein, we describe the operation mechanism of the rectifier circuit. When the input alternative current (AC) signal is negative, the capacitor *C*_1_ is charged through the diode *D*_1_. When the input is positive, *D*_1_ is turned off and *D*_2_ is turned on; thereby, *C*_2_ is charged. Thus, we may obtain a DC output voltage with a level of the AC magnitude by the repeatedly charging process. Both *C*_1_ and *C*_2_ in Figure 4a are 100 pF. Schottky diodes (SMS7630-061) with low threshold voltage (0.34 V) are chosen as *D*_1_ and *D*_2_ due to the low amplitude of the RF signal received from the WiFi antenna. Owing to manufacture processing limitations, the actual thickness of FR4 substrates (*t*_2_) is 1.6 mm in the fabricated MIA. In order to reflect the loss in the FR4 substrate, the dielectric constant of FR4 is 3.4 and the tangent loss is 0.022 in the system level simulation.

The wireless energy-harvesting capability of the system is evaluated by utilizing the system-level simulation model, as shown in Figure 4b–g. The rectified voltage peaks at the resonant frequency of the MIA, which is approximately 2.425 GHz, and the voltage is sensitive to the frequency of the input signal. When the transmit power is less than 0 dBm, the maximum voltage output is less than 150 mV. A high transmit power is required to ensure that the MIA rectifies sufficient voltages. As the distance between the WiFi transmit antenna and the MIA becomes larger, the DC voltage output by the rectifier circuit under the condition of different transmit powers gradually decreases. When the distance is 2 cm and transmit power is 9 dBm, the maximum output DC voltage is about 525.6 mV. When the distance is 10 cm and the transmit power is 9 dBm, the output DC voltage is ≈24.45 mV.

## 3. Results and Discussion

### 3.1. Open-Circuit Output Characteristics

We fabricated the MIA using the standard printed circuit board (PCB) process and soldered the rectifier circuit. Figure 5a shows a photograph of an experimental setup for the output voltage of a WiFi energy-harvesting MIA. A commercial WiFi antenna was connected to the output port of a vector network analyzer (Agilent E8361A) by a co-axial cable. The RF electromagnetic signal was emanated from the WiFi antenna, and its operating frequency can be tuned by configuring the vector network analyzer. The wireless energy-harvesting MIA was attached on a 3D-printed holder. The MIA was placed at a fixed distance from the WiFi antenna, and the rectified DC voltage at the MIA was measured using an oscilloscope. The type of digital oscilloscope we used is Tektronix TBS2000B, and its input impedance is 1 MΩ. The size of the MIA PCB board is 14 mm × 14 mm.

To confirm the resonance frequency of the proposed MIA harvester, we investigated its open-circuit output voltages by changing the operating frequencies in the range from 2 to 2.8 GHz with a step of 0.025 GHz. The distance between the MIA and the WiFi antenna was 2 cm. The measurement results and corresponding simulation results are plotted in Figure 5c. According to the experimental results, the output voltage reaches its maximum value (≈500 mV) at a frequency of 2.425 GHz. The good agreement between the experiment and simulation validates the system-level simulation.

In order to further characterize the energy-harvesting performance of the MIA, the rectified output voltage of circuits was measured at different distances and input RF powers. The distance between the WiFi antenna and the MIA harvester was changed from 2 to 10 cm with a step of 1 cm. The delivered power of the RF signal (2.425 GHz) was increased from −2 dBm (0.6 mW) to 9 dBm (8 mW) with a step of 1 dBm. As shown in Figure 5d, the output voltage increases as the input RF power increases. At a distance of 2 cm, the maximum output DC voltage can reach 494 mV at an input RF power of 9 dBm. In addition, the output voltage is greater than 100 mV at an input RF power of −1 dBm. The MIA harvester does not affect the normal operation of the WiFi antenna due to its small size.

Figure 5e shows the rectified output voltage as a function of distance. As expected, the rectified voltage drops as the distance increases. At a moderate distance of 6 cm, a DC voltage of 216 mV can be achieved at an input RF power of 9 dBm. The working distance of the MIA can also be further increased if the transmit power increases.

### 3.2. Load Effect of the MIA Harvester

After measuring the open-circuit voltage of the MIA harvester, various resistive loads, including 1 KΩ, 3 KΩ, 10 KΩ and 30 KΩ, were added to the output port of the rectifier circuit. The experimental setup shown in Figure 5a was also used to measure the voltage on the load. Herein, the power of the transmitted RF signal was 9 dBm, and the frequency was set to 2.425 GHz. As depicted in Figure 6a, the output DC voltage decreases from 225 to 9 mV as the distance increases from 2 to 10 cm. Among the four different load resistances, a DC voltage of about 500 mV can be reached if the load resistance was 10 KΩ and 30 KΩ. As shown in Figure 6d, the output DC voltage is more than 100 mV if the distance is 10 cm. In addition, the calculated load current can be up to 0.2 mA if the resistance was 1 KΩ. In order to reduce the measurement error, we have measured four times at each position and calculated their average as measurement results.

We calculated the output DC power ($P_{load}$), which is defined as $P_{load}=V_{out}^{2}/R_{load}$, where $V_{out}$ is the output DC voltage and $R_{load}$ is the load resistance. According to the measured results shown in Figure 6, the MIA harvester demonstrated the capability of harvesting the RF power in the WiFi band (2.425 GHz) and generated a rectified output power of up to 50 μW for the 1 KΩ resistance. It should be noted that this level of output power is able to support the operation of the microsystem [58,59]. The output power of the 30 KΩ resistor was lower than that of the 1 KΩ resistor because of the large resistance. The performance of the presented MIA harvester in this paper is compared with previously published works, as shown in Table 2. Based on the data presented in the table, the proposed MIA is small and capable of collecting WiFi signals at 2.425 GHz for self-powered microsystems and miniaturized IOT nodes.

The output voltage and power of the MIA harvester can be improved by increasing the stages of the voltage multiplier in the rectifier circuit and optimized for specific load. A Dickson voltage multiplier with three stages is demonstrated in Figure 7a. The voltages in the cascaded structure of diode capacitors are not reset after each pumping cycle. Each stage utilizes the voltage of its previous stage as a reference. As a result, the DC output of a multi-stage Dickson voltage multiplier can be several times of the input amplitude. When the distance is 2 cm and transmit power is 9 dBm, the maximum output voltage is up to 2.62 V by a three-stage Dickson voltage multiplier, as shown in Figure 7b. A DC voltage of this magnitude may be sufficient to power most electronic devices and systems. Moreover, an increased transmit power may improve the working distance of MIA in practical applications.

## 4. Conclusions

We have developed a miniaturized WiFi energy harvester inspired by the resonant magnetic dipole metamaterial. The MIA consists of two identical square SRRs, including a resonant magnetic dipole and a coupling coil. The side length of two SRRs is 10.5 mm, which is less than 0.1 λ. Under the excitation of the EM wave, an oscillating current will be induced in the resonant magnetic dipole. The impedance matching can be achieved by changing the thickness of the FR4 substrate to tune the near-field coupling of the SRRs. An analytical ECM was built to unveil the effect of the near-field interaction. Therefore, we provide a simple and effective solution for impedance matching instead of complicated matching networks. In addition, the ability to harvest WiFi energy was studied using numerical simulation and verified experimentally. At a distance of 2 cm from the WiFi transmit antenna (9 dBm), an output DC voltage of approximately 0.5 V can be measured after being processed by a single-stage Dickson voltage multiplier. The MIA proposed in this paper paves a pathway to power microsystems for many potential applications.

## Figures and Tables

**Figure 1 sensors-22-06523-f001:**
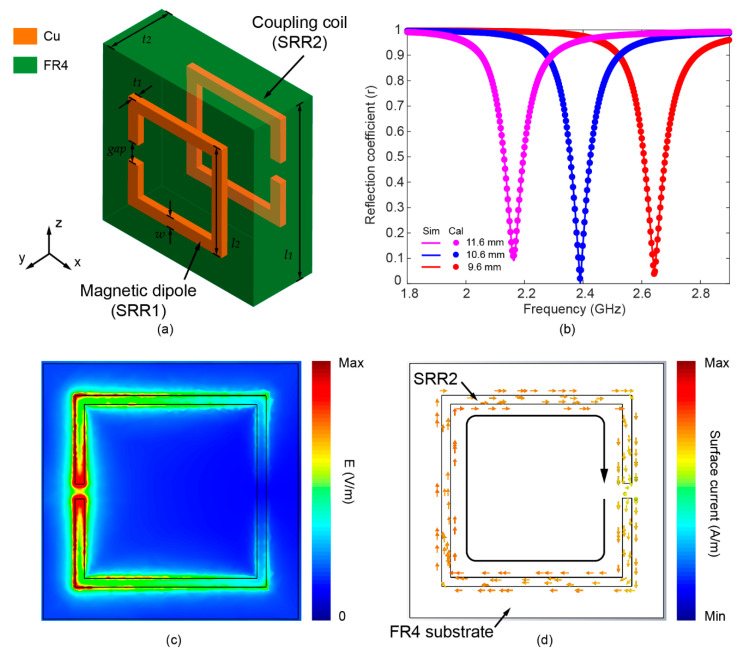
(**a**) The illustration of the presented metamaterial-inspired antenna (MIA). (**b**) Simulated and calculated reflection coefficients of the MIA with various lengths of SRRs (*l*_2_). (**c**) The simulated electric field distribution in the plane of resonant magnetic dipole (SRR1). (**d**) The simulated surface current in the plane of coupling coil (SRR2).

**Figure 2 sensors-22-06523-f002:**
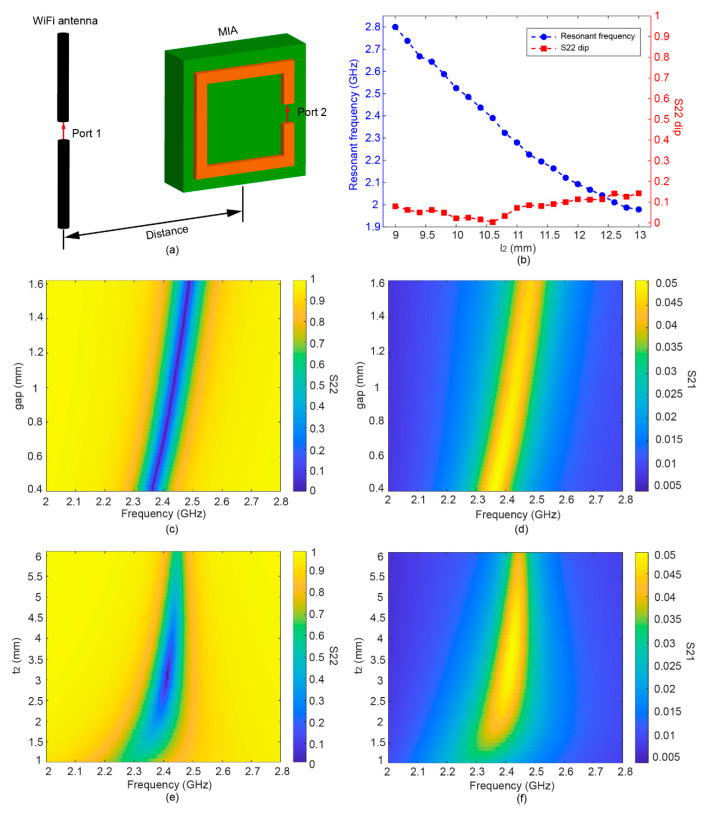
(**a**) The simulation model to analyze the RF characteristics of the MIA. (**b**) Simulated minimum of the reflection coefficient of the MIA, i.e., minimum of S22, and resonant frequency of the MIA versus length of SRR (*l*_2_). (**c**) Simulated S22 and (**d**) transmission coefficient (S21) versus the gap of SRRs (*gap*). (**e**) Simulated S22 and (**f**) S21 versus thickness of FR4 substrate (*t*_2_).

**Figure 3 sensors-22-06523-f003:**
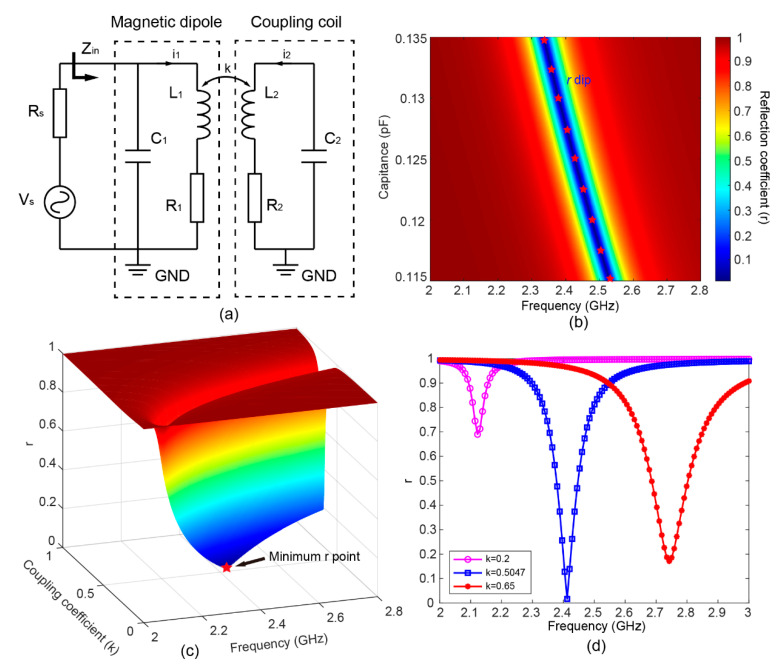
(**a**) Lumped circuit model of the MIA. (**b**) Calculated reflection coefficient (*r*) versus capacitances and frequency. (**c**) Calculated *r* values versus coupling coefficient and frequency. (**d**) The reflection spectrum with three coupling coefficient values are selected and plotted (*k* = 0.2, *k* = 0.5047, and *k* = 0.65). The calculated *k* of the optimized MIA is 0.5047 and its resonant frequency is 2.413 GHz.

**Figure 4 sensors-22-06523-f004:**
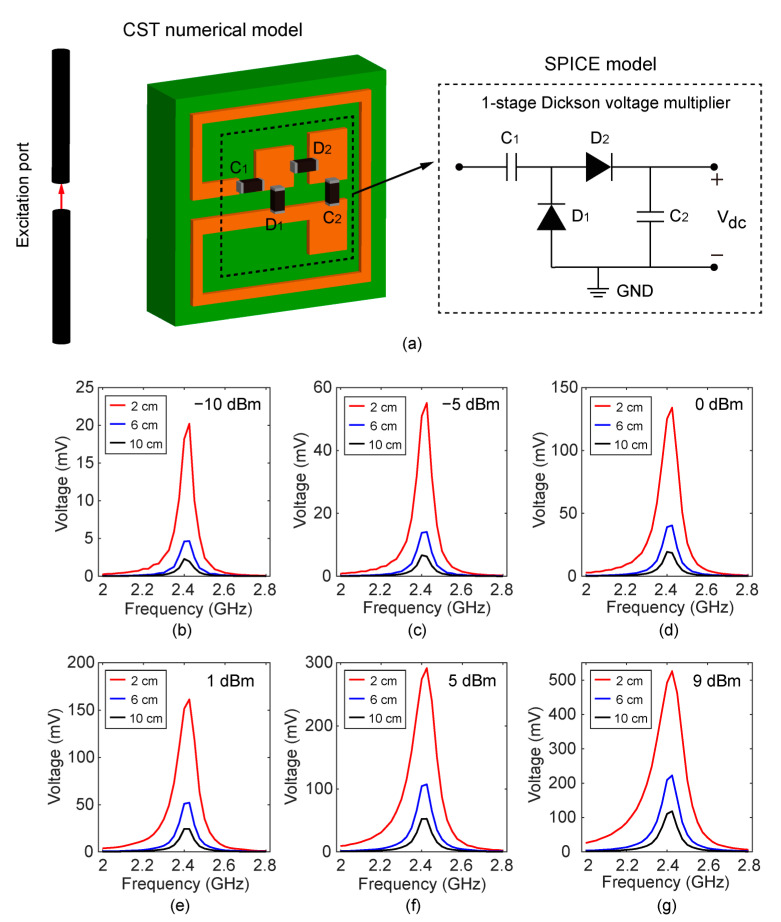
(**a**) The system-level simulation model to analyze the wireless energy-harvesting capability of MIA. Simulated output DC voltage (open circuit) versus frequency under different transmitting powers: (**b**) −10 dBm, (**c**) −5 dBm, (**d**) 0 dBm, (**e**) 1 dBm, (**f**) 5 dBm, and (**g**) 9 dBm.

**Figure 5 sensors-22-06523-f005:**
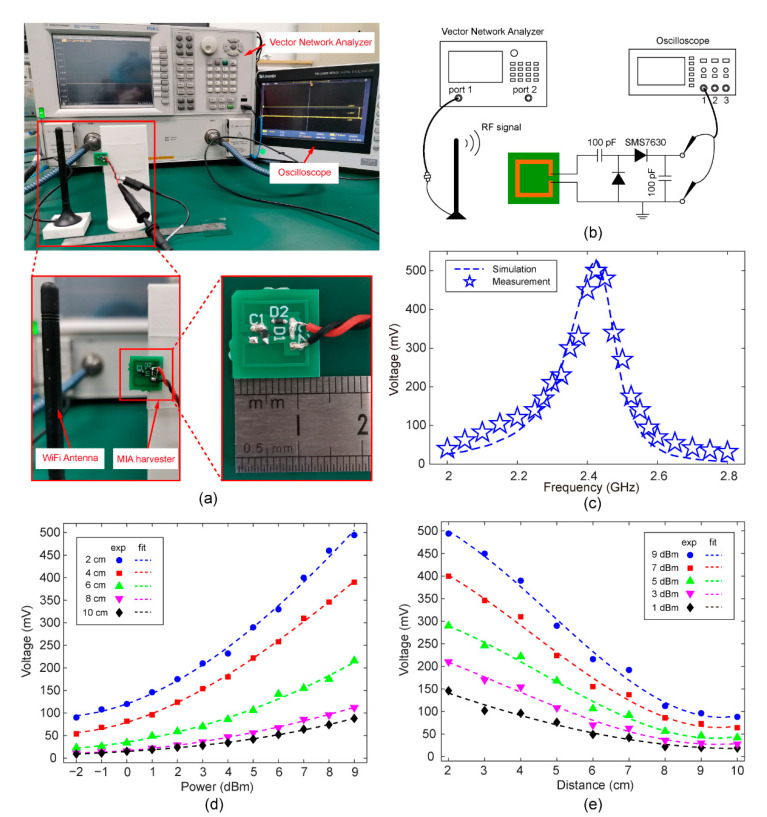
(**a**) Experimental setup for the output voltage of WiFi energy-harvesting MIA. (**b**) Schematic diagram of the experimental setup. (**c**) Simulated (blue dotted line) and measured DC output voltage spectrum (blue pentagram) of the MIA harvester. (**d**) The relationship between measured output voltages and input RF powers for different distances. (**e**) The relationship between measured output voltages and distance under conditions of different input RF powers. The experimental results and polynomial fitting results are represented by dotted lines and markers, respectively.

**Figure 6 sensors-22-06523-f006:**
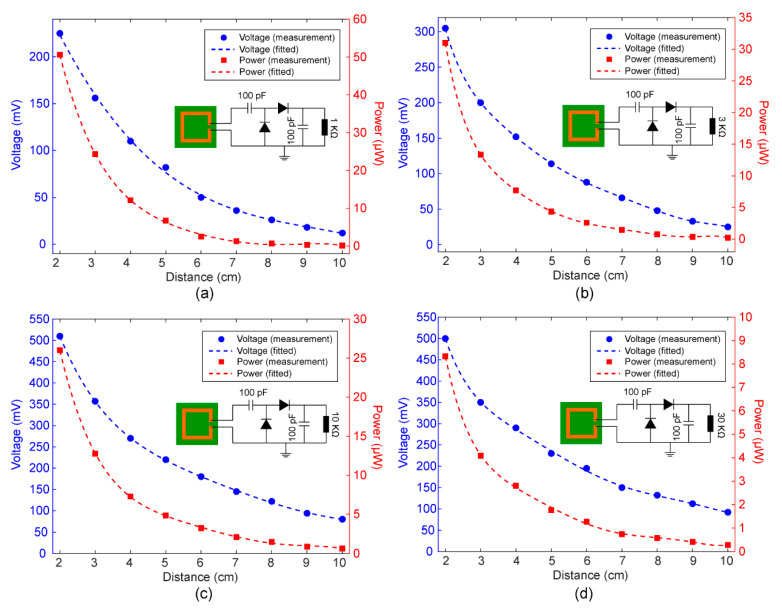
Measured output DC voltages and calculated output powers with different load resistances at 2.425 GHz. (**a**) 1 KΩ. (**b**) 3 KΩ. (**c**) 10 KΩ. (**d**) 30 KΩ. The inset depicted the detailed schematic diagram of PCB circuits with various resistors.

**Figure 7 sensors-22-06523-f007:**
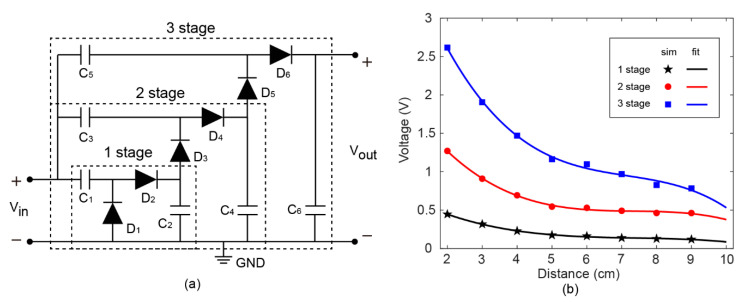
(**a**) The schematic of the N-stage Dickson voltage multiplier (N = 1, 2, 3). (**b**) Simulated output DC voltage (open circuit) versus distance under different stages of Dickson voltage multiplier.

**Table 1 sensors-22-06523-t001:** Design parameters of the optimal MIA.

Parameters	Symbol	Value
Length of PCB board	*l* _1_	14 mm
Length of SRR	*l* _2_	10.5 mm
Line width of SRR	*w*	0.5 mm
Gap of SRR	*gap*	0.8 mm
Thickness of copper wire	*t* _1_	50 μm
Thickness of FR4 substrate	*t* _2_	3.1 mm

**Table 2 sensors-22-06523-t002:** Performances for various wireless energy harvester in the RF band.

Authors	Frequency	Size	Output Voltage (Transmit Power/Distance)
Hawkes et al. [40]	900 MHz	>1600 mm^2^ (5 × 1)	<2.5 V (15 dBm/NA)
Zhang et al. [21]	5.9 GHz	>1900 mm^2^	250 mV (3 dBm/2.5 cm)
Sun et al. [23]	2.45 GHz	6960 mm^2^	<2.5 V (5 dBm/NA)
Lee et al. [41]	2.4 GHz	>300 mm^2^	2 V (Mobile Hotpot/5 cm)
Chandravanshi et al. [24]	2.45 GHz	6440 mm^2^	0.98 V (20 dBm/20 cm)
This work	2.425 GHz	196 mm^2^	0.494 V (9 dBm/2 cm)

## Data Availability

Not applicable.

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
