# Peer review of "WiFi Energy-Harvesting Antenna Inspired by the Resonant Magnetic Dipole Metamaterial"

_sensors, 2022, doi:10.3390/s22176523_

Round 1

Reviewer 1 Report

The authors presented a metamaterial-inspired antenna (MIA) based on 18 the resonant magnetic dipole operating in the WiFi bands. The work is well presented and validated with good circuit analysis, simulation, and prototype characterization. The work holds good scientific value and adds value to the existing literature of energy harvesting. This reviewer is recommending to accept the manuscript after addressing the following comments:

1. Though the prototype was characterized with 1-stage rectifier stage, this reviewer is recommending to include simulation results with increasing the stages of Dickson charge pump.

2. Also, please include system-level simulation results of the energy harvester with some small signal power, e.g., -10 dBm, -5 dBm, 0 dBm.

Author Response

Dear Reviewers,

We would like to thank you for the valuable comments and suggestions regarding our manuscript entitled “WiFi Energy-Harvesting Antenna Inspired by the Resonant Magnetic Dipole Metamaterial”. We find that all of the comments and suggestions are constructive and helpful. We have attempted to address all of the comments and concerns. Please see the attachment.

Reviewer 2 Report

1. What is the practicality of harvesting energy at 2 cm distance from Wifi antenna? If the antenna is placed near to the WiFi antenna, will it affecting its normal operation? 

2. What is the performance of the harvester more than a meter? 

3. Please comment if multiple stages of voltage multiplier to get higher output voltage is also possible. 

4. What is the load resistance / current at 0.5 V output?

5. In Figure 5, what is the oscilloscope's input impedance in measuring the output voltage of the voltage multiplier.

6. The paper is also lack of comparison with existing work. 

Author Response

(The authors gave the same response as above.)

Reviewer 3 Report

This work reports a metamaterial-inspired antenna (MIA) based on the resonant magnetic dipole which is made of two metallic split-ring resonators (SRRs), separated by an FR4 dielectric layer, in the broadside coupled configuration. By varying the vertical distance of the two SRRs, impedance matching without complicated matching networks can be achieved. The designed MIA resonates at 2.4 GHz and a rectified DC voltage of 500 mV is obtained when the MIA is placed at a distance of 2 cm from the WiFi transmit antenna with a 9-dBm transmitting power. The manuscript is well written and data presented is reasonable, so I think it can be published after minor revisions as below:

1.     There are two abbreviations of S22 in line 122 and line 130, while there is no explanation for what is S21 across the paper. Please check it.

2.     In table 1, l2 represents the length of SRR, while in line 156, l is the total length of the SRR. Please clarify the difference between l2 and l.

Author Response

(The authors gave the same response as above.)
